

# RNA sequencing reveals the emerging role of bronchoalveolar lavage fluid exosome lncRNAs in acute lung injury

Meijuan Song[1,*], Xiuwei Zhang[2,*], Yizhou Gao[3], Bing Wan[2], Jinqiang Wang[4], Jinghang Li[3], Yuanyuan Song[3], Xiaowei Shen[3], Li Wang[2], Mao Huang[1] and Xiaowei Wang[3]

[1] Department of Respiratory and Critical Care Medicine, The First Affiliated Hospital with Nanjing Medical University, Nanjing, Jiangsu, China

[2] Department of Respiratory and Critical Care Medicine, The Affiliated Jiangning Hospital of Nanjing Medical University, Nanjing, Jiangsu, China

[3] Department of Cardiovascular Surgery, The First Affiliated Hospital with Nanjing Medical University, Nanjing, Jiangsu, China

[4] Department of Intensive Care Unit, Xuchang People's Hospital, Xuchang, Henan, China

* These authors contributed equally to this work.

## ABSTRACT

**Background:** Bronchoalveolar lavage fluid (BALF) exosomes possess different properties in different diseases, which are mediated through microRNAs (miRNAs) and long noncoding RNAs (lncRNAs), among others. By sequencing the differentially expressed lncRNAs in BALF exosomes, we seek potential targets for the diagnosis and treatment of acute lung injury (ALI).

**Methods:** Considering that human and rat genes are about 80% similar, ALI was induced using lipopolysaccharide in six male Wistar rats, with six rats as control (all weighing 200 ± 20 g and aged 6–8 weeks). BALF exosomes were obtained 24 h after ALI. The exosomes in BALF were extracted by ultracentrifugation. The differential expression of BALF exosomal lncRNAs in BALF was analyzed by RNA sequencing. Gene Ontology (GO) and Kyoto Encyclopedia of Genes and Genomes (KEGG) analyses were performed to predict the functions of differentially expressed lncRNAs, which were confirmed by reverse transcription–polymerase chain reaction.

**Results:** Compared with the control group, the ALI group displayed a higher wet/dry ratio, tumor necrosis factor-α levels, and interleukin-6 levels (all $P < 0.001$). The airway injection of exosomes in rats led to significant infiltration by neutrophils. A total of 2,958 differentially expressed exosomal lncRNAs were identified, including 2,524 upregulated and 434 downregulated ones. Five lncRNAs confirmed the reliability of the sequencing data. The top three GO functions were phagocytic vesicle membrane, regulation of receptor biosynthesis process, and I-SMAD binding. Salmonella infection, Toll-like receptor signaling pathway, and osteoclast differentiation were the most enriched KEGG pathways. The lncRNA–miRNA interaction network of the five confirmed lncRNAs could be predicted using miRDB.

**Conclusions:** BALF-derived exosomes play an important role in ALI development and help identify potential therapeutic targets related to ALI.

Corresponding authors
Mao Huang, hm6114@163.com
Xiaowei Wang, wangxiaowei@njmu.edu.cn

# INTRODUCTION

Acute lung injury (ALI) is a severe respiratory condition characterized by pulmonary vascular endothelial and epithelial cell damage, leading to diffuse interstitial edema and alveolar edema. The pulmonary shunt can be caused by a larger dead space of the lungs due to alveolar edema (*Confalonieri, Salton & Fabiano, 2017*). These events impair lung compliance and oxygen exchange, leading to acute respiratory dysfunction and high mortality (30–40%) (*Bellani et al., 2016*). ALI is diagnosed mostly based on clinical features (*Johnson & Matthay, 2010*; *Labib et al., 2018*). Identifying biomarkers for diagnosing ALI and monitoring the effects of treatments are of significance. Almost all cells secrete extracellular vesicles (EVs). These vesicles contain proteins, lipids, and nucleic acids that are passed from the mother cell to the recipient cell. Therefore, they act as a medium for cell-to-cell communication and molecular transfer.

Microvesicles (MVs), apoptotic bodies, and exosomes are grouped as EVs (*El Andaloussi et al., 2013*). Exosomes are considered as the miniature versions of parental cells because not only they have the same lipid bilayer as donor cells and carry rich proteins, DNA, lipids, and RNA from donor cells, but also their functions are closely related and can reflect the characteristics of parental cells (*Daaboul et al., 2016*; *Llorente et al., 2013*; *Pitt, Kroemer & Zitvogel, 2016*; *Skotland, Sandvig & Llorente, 2017*; *Théry, Zitvogel & Amigorena, 2002*; *Théry, Ostrowski & Segura, 2009*). Exosomes are found in almost all biological fluids (*Torregrosa Paredes et al., 2012*; *Yang et al., 2019*). Human bronchoalveolar lavage fluid (BALF) contains exosomes displaying the major histocompatibility complex class II and co-stimulatory molecules (*Admyre et al., 2003*). Phenotypic and functional differences in BALF exosomes exist between asthmatic and healthy individuals (*Martin-Medina et al., 2018*; *Torregrosa Paredes et al., 2012*). *Torregrosa Paredes et al. (2012)* found that the BALF exosomes from asthmatic patients could promote subclinical inflammation *via* increasing cytokine and leukotriene C production by the airway epithelium. In addition, elevated numbers of BALF EVs (especially exosomes) are observed in patients with idiopathic pulmonary fibrosis; the production of the pro-fibrotic growth factor-β through the WNT5A signaling pathway can be induced by these EVs, promoting the progression of fibrosis (*Martin-Medina et al., 2018*). Nevertheless, exosomes carry various molecules, and some can have beneficial effects. Indeed, microRNA (miRNA)-26 can be delivered from human endothelial progenitor cells to injured alveoli by exosomes, reducing ALI-related inflammation and improving prognosis (*Zhou et al., 2019*). Macrophages secrete many early pro-inflammatory cytokines in BALF exosomes, and these exosomes contribute to neutrophil activation and the secretion of pro-inflammatory cytokines and IL-10 (*Ye et al., 2020*).

Long noncoding RNAs (lncRNAs), miRNAs, proteins, metabolites, and other substances can deliver vital information to various cells through exosomes (*Bovy et al., 2015*;

*Fujita et al., 2015*; *Njock et al., 2015*; *Pua et al., 2019*; *Xu et al., 2018*). Also, miRNAs and lncRNAs from exosomes can be used as biomarkers, treatment guides, and mechanistic markers for the pathogenesis and progression of ALI (*Lee et al., 2019*). This role of exosomal miRNAs and lncRNAs has been proven in tumor growth, metastasis, and angiogenesis (*Lin & Yang, 2018*; *Zhao et al., 2019*). *Chen et al. (2020)* found that monocyte-derived exosomal lncRNA (CLMAT3) could activate the 85 CtBP2–p300–NF-κB transcription complex to induce pro-inflammatory cytokines in ALI. *Mohamed Gamal El-Din et al. (2020)* showed the use of lnc-RNA-RP11-510M2.10 to diagnose and determine the prognosis of lung cancer. LncRNAs are also involved in acute brain and kidney injury (*Brandenburger et al., 2018*; *Chandran, Mehta & Vemuganti, 2017*), but the data on exosome lncRNA serving as a target for the diagnosis and treatment in ALI are still lacking.

BALF is a common body fluid used for the diagnosis of lung diseases. It more directly reflects the lung tissues and cells compared with blood (*Chang et al., 2020*). This study aimed to identify differentially expressed genes in BALF exosomes by RNA sequencing and suggest potential therapeutic targets of ALI. Considering that human and rat genes are about 80% similar (*Zhao et al., 2004*), rats were used in the present study.

## MATERIALS AND METHODS

### Animals

The experiments were performed adhering to the institutional guidelines and approved protocols. The animal experiments were approved by the Institutional Animal Care and Use Committee of Nanjing Medical University (No. IACUC-2004021). All animal experiments were conducted at the Animal Core Facility of Nanjing Medical University. Twelve male Wistar rats (weighing 200 ± 20 g and aged 6–8 weeks) were purchased from Nanjing Qing Long Shan animal farm (Nanjing, China).

During the whole experiment, the rats in the control and the experimental groups had free access to food and water. The health and immune statuses of all rats used were normal, and they were not involved in any previous procedures. The rats were randomly grouped (random number table method) as ALI models and controls ($n = 6$ per group). ALI was modelled and the sample size was determined as previously described (*Do-Umehara et al., 2013*; *Lu et al., 2012*). Lipopolysaccharide (LPS) was dissolved in 0.5 mL of normal saline to obtain a solution at 10 mg/kg of body weight. After anesthesia with 3% sodium pentobarbital (50 mg/kg), the rats were placed in a supine position on the operating table and airway-injected with the LPS solution. An equal volume of normal saline was given to the rats in the control group. All Wistar rats were placed under the same conditions for 24 h and given the same anesthesia. All animals were given humane care.

For confirming the properties of the exosomes, the exosomes purified from the ALI group were resuspended in 200 μL of phosphate-buffered saline (PBS) and infused into the lungs of two rats. The rats in the control group was infused with PBS alone. The histological examination was performed 24 h later. Only investigator who performed modeling was aware of grouping but was not involved in the subsequent experiments or analyses.

## BALF sampling and histopathological analysis

Twelve Wistar rats were divided into the experimental ($n = 6$) and control ($n = 6$) groups. Anesthesia was performed with 3% pentobarbital sodium (50 mg/kg). After successful confirmation of endotracheal intubation using alc-8 small-animal ventilator, 5 mL of normal saline (0.9%) was injected into the airway. Through airway intubation, the right lung was ligated and the left lung was irrigated with 4 °C pre-cooled saline. This was repeated four times, and the BALF was collected in centrifuge tubes. Once the BALF was obtained, the rats were sacrificed by cervical dislocation and the lungs were harvested. The left lung was weighed (wet weight), placed in an oven at 65 °C for 7 d, and then weighed again to determine the dry weight. The dry-to-wet ratio was calculated. The right lung was formalin-fixed and paraffin-embedded. The sections (4 μm) were cut and stained with hematoxylin and eosin.

## Enzyme-linked immunosorbent assay

Commercial enzyme-linked immunosorbent assay were used to measure the levels of interleukin (IL)-6 and tumor necrosis factor (TNF)-α from 12 rats ($n = 6$/group) (R&D Systems, Minneapolis, MN, USA) following the manufacturer's protocol.

## Extraction of exosomes from BALF

The BALF exosomes were purified following the ISEV guidelines (*Deady et al., 2014*; *Théry et al., 2018*). This includes determining the speed of ultracentrifugation based on rotor type, tube/adapter, and centrifuge speed. Second, the pore size of the matrix should be considered. For example, a group of vesicles may be excluded if the pore size does not include EVs >70 nm in diameter. As well as EV characterization based on protein content, at least one of 1a (CD63, CD81, CD82, etc.) or 1b (ERBB2, EPCAM, CD90, etc.), 2a (TSG101, HSPA8), 3a (APOA1/2, APOB; APOB100, etc.) or 3b (Tamm-Horsfall protein, UMOD) class proteins must be analysed to demonstrate the properties of EVs and the purity of EV preparations. For this, 15 mL of BALF samples were centrifuged for 10 min at 2,000$g$. The supernatant was centrifuged for 20 min at 12,000$g$ (Optima L-100XP Ultracentrifuge, Beckman Coulter, Brea, CA, USA). The supernatant was centrifuged again for 70 min. After centrifugation, the supernatant was discarded, and the precipitate (exosomes) was resuspended in 200 μL of PBS in a 1.5-ml eppendorf tube and stored at −80 °C.

## Exosome properties

A Tecnai G2 Spirit BioTwin Nano Transmission Electron Microscope (FEI, Hillsboro, OR, USA) detector was used to examine the exosome morphology. A nanoparticle size detector was used to detect exosome particle size.

## Western blotting

Exosome surface proteins (CD63 and TSG101) were examined by Western blot (*Keerthikumar et al., 2016*; *Lee, El Andaloussi & Wood, 2012*; *Logozzi et al., 2009*). Equal amounts of proteins from the samples were resolved by sodium dodecyl

sulfate–polyacrylamide gel electrophoresis and transferred on to a poly-polarized PVDF membrane. The blot was incubated with the primary antibodies overnight: mouse anti-GAPDH (1:5,000; ab8245; Abcam, Cambridge, United Kingdom), mouse anti-CD63 (1:1,000; sc-5275; Santa Cruz Biotechnology, Santa Cruz, CA, USA), and rabbit anti-Tsg101 (1:5,000; ab125011; Abcam, Cambridge, United Kingdom). The secondary antibody was HRP-conjugated goat anti-rabbit IgG or goat anti-mouse IgG (both 1:50,000; Wuhan Boster Biological Technology, Ltd., Wuhan, China). The bands were revealed using an ECL reagent (Pierce Chemical, Dallas, TX, USA). The film was scanned and analysed using BandScan.

## RNA-seq

Four samples were randomly selected from the two sets of samples for high-throughput transcriptome sequencing. We carried out quality inspection on the sample RNA, and explained the detection index RIN (RNA Integrity Number) of RNA integrity. RIN ranges from 0 to 10. The higher the score, the better the integrity of the RNA. The RNA quality inspection result of our sample is >=7.0, which is a qualified sequencing sample, and the base distribution was balanced. For raw reads that might contain unqualified reads with low overall quality, sequencing primers, low end quality, and so forth, we applied Seqtk (https://github.com/lh3/seqtk) to filter them to obtain clean reads that could be used for data analysis. The RNeasy mini kit (Qiagen, Venlo, The Netherlands) was used to isolate total RNA. The TruSeq RNA Sample Preparation Kit (Illumina, Inc., San Diego, CA, USA) was used to synthesize paired-end libraries. The poly-A-containing mRNA molecules were purified using poly-T oligo-attached magnetic beads. A Qubit 2.0 Fluorometer (Life Technologies Co., Grand Island, NY, USA) was used to quantify the purified libraries, which were validated using an Agilent 2100 bioanalyzer (Agilent Technologies, Santa Clara, CA, USA). The cluster was generated using cBot with the library diluted to 10 pM. The cluster was sequenced on an Illumina HiSeq X-ten (Illumina, Inc., San Diego, CA, USA). Shanghai Biotechnology Corporation (Shanghai, China) performed library construction and sequencing.

Unqualified reads were filtered to obtain clean reads for data analysis using Seqtk (https://github.com/lh3/seqtk) for filtering (version 2.2.8). The reads were preprocessed by filtering out rRNA reads, sequencing adapters, short-fragment reads, and other low-quality reads using Hisat2 (version 2.0.4) (*Kim, Langmead & Salzberg, 2015*) to map the cleaned reads to the human GRCh38 reference genome with two mismatches. The novel lncRNA and NONCODE database (version: NONCODE 2016; http://www.noncode.org/) were predicted using Stringtie (version:1.3.0) (*Pertea et al., 2015*, *2016*), and the known data in the Ensembl database lncRNA were used for expression quantification. The ID starting with MSTRG was novel lncRNA, the ID starting with NON was the known lncRNA in the database, and the ID starting with ENS was the known lncRNA in the Ensembl database.

Stringtie (version 1.3.0) was run with a reference annotation to generate fragments per kilobase of exon model per million mapped reads (FPKM) values for known gene models. Differentially expressed genes were identified using edgeR

(*Robinson, McCarthy & Smyth, 2010*). The *P* value was set using the false discovery rate (FDR) (*Benjamini et al., 2001*; *Benjamini & Hochberg, 1995*; *Benjamini & Yekutieli, 2001*). The fold-changes were also estimated according to the FPKM in each sample. The differentially expressed genes were selected using the following filtering criteria: FDR ≤0.05 and fold-change (FC) 195 ≥ 2.

StringTie (*Pertea et al., 2015*, *2016*) (version: 1.3.0) was applied to quantify the expression of novel lncRNAs and NONCODE databases predicted using 2.2.11 (version: NONCODE 2016; http://www.noncode.org/), as well as examine known lncRNAs in Ensemble database.

### GO and KEGG analysis of differentially expressed lncRNAs
The reads were converted into FPKM for standardized gene expression levels (*Mortazavi et al., 2008*; *Robinson, McCarthy & Smyth, 2010*) for comparisons between groups. The differentially expressed lncRNAs were used for Gene Ontology (GO) enrichment analysis (http://geneontology.org/) and Kyoto Encyclopedia of Genes and Genomes (KEGG) pathway enrichment (http://www.genome.jp/kegg). All experiments were performed three times independently. Subsequently, five differentially expressed lncRNAs were randomly selected for validation.

### Real-time quantitative reverse transcription–polymerase chain reaction
The miRNeasy Micro Kit kit (Qiagen, Venlo, The Netherlands) was used to extract exosomal total RNA. Table 1 presents the primers for quantitative polymerase chain reaction (qPCR). The amplification parameters were 95 °C for 10 s and 60 °C for 34 s, for a total of 40 cycles. The relative expression levels of lncRNAs were calculated using the $2^{-\Delta\Delta Ct}$ method (*Livak & Schmittgen, 2001*).

### Prediction of the lncRNA–miRNA interaction networks
Five lncRNAs were selected to construct lncRNA–miRNA networks using the miRDB Database to investigate the regulation network between lncRNAs and their target miRNAs. The Cytoscape software (version 3.7.1, https://cytoscape.org/) was used for network visual representation.

### Statistical analysis
Data were tested for normal distribution using the Shapiro–Wilk test. They were presented as means ± standard deviations and analyzed using Student *t* test. All analyses were performed using SPSS 17.0 (IBM, New York, NY, USA). *P* values < 0.05 indicated statistically significant differences. The rats that did not meet the ALI standard were excluded.

## RESULTS

### ALI modeling and the pro-inflammatory effects of exosomes
Compared with the control group (*n* = 6), the lung tissues in the ALI group (*n* = 6) showed a significantly smaller alveolar cavity, more extensive alveolar space, and infiltration of

**Table 1 The primers used in qPCR.**

| Primers | Sequence | Product length |
|---|---|---|
| NONRATT002967.2 (forward)<br>NONRATT002967.2 (reverse) | ACTTTACAAGCCGGAGGACG<br>GAGTTGGGAGCGTTTGGAGA | 117 bp |
| NONRATT003362.2 (forward)<br>NONRATT003362.2 (reverse) | ATCCACTTCTGTCTGAGGGC<br>GGAAGGTGCGTTGAACACTT | 108 bp |
| NONRATT004060.2 (forward)<br>NONRATT004060.2 (reverse) | ACAGCCAGATCGCCAGTAAA<br>GAAGGCTCCAATCTGCTCTGT | 161 bp |
| NONRATT025040.2 (forward)<br>NONRATT025040.2 (reverse) | TTGCTCCTCGACTCTTCGTG<br>CGGAGAGCGTAGACTCGGAA | 145 bp |
| NONRATT025699.2 (forward)<br>NONRATT025699.2 (reverse) | GGATACTAAAGCAGCCTTGCAC<br>CACCTCCACAGCAAAGCTTAC | 165 bp |

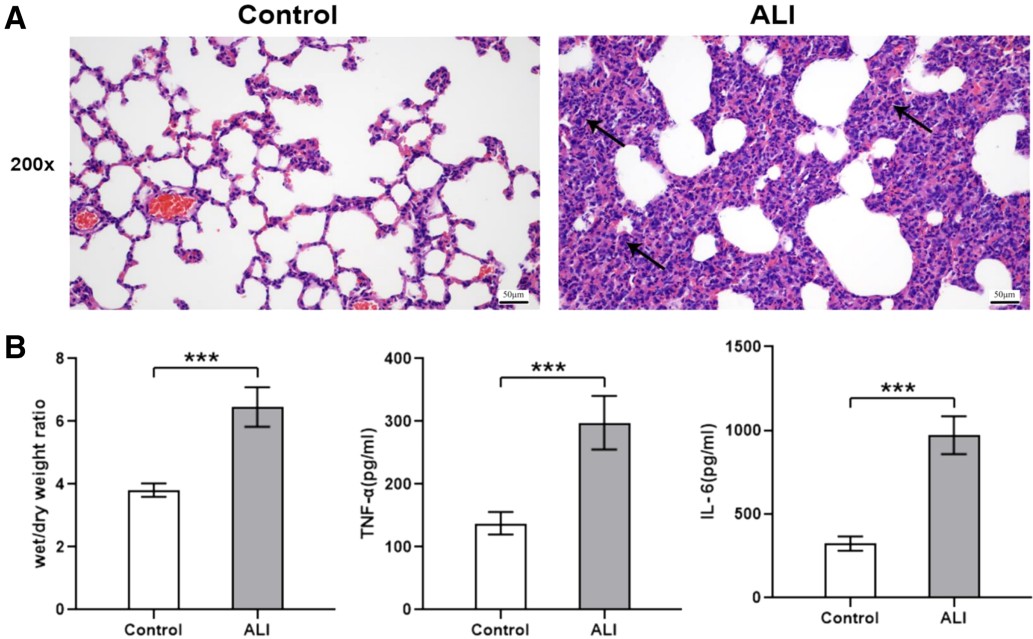

**Figure 1 ALI induced histological changes, and ALI exosomes were involved in the inflammatory response.** Twelve male Wistar rats were modeled (weighing 200 ± 20 g and aged 6–8 weeks), six each in the ALI and control groups. After 24 h of modeling, the right lung was ligated and the left lung was subjected to bronchoalveolar lavage fluid (BALF) extraction. The lungs were harvested afterward. (A) Lung histopathological examination of rats in control and LPS-induced ALI groups ($n = 4$/group). Filtration of a large number of inflammatory cells was seen and some lung tissue structures were destroyed in the ALI group compared with the control group. Scale bar: 200×. (B) Wet/dry weight ratio and BALF TNF-α and IL-6 levels were determined. Enzyme-linked immunosorbent assay data are representative of three independent experiments ($n = 6$). ***$P < 0.001$ *versus* the control group.

many neutrophils in the alveolar wall (Fig. 1A). Compared with the control group, the ALI group showed a higher wet/dry ratio, TNF-α levels, and IL-6 levels (all $P < 0.001$). These results indicated that exosomes were involved in mediating inflammatory responses in ALI (Fig. 1B). In addition, our study found that the airway injection of exosomes in rats
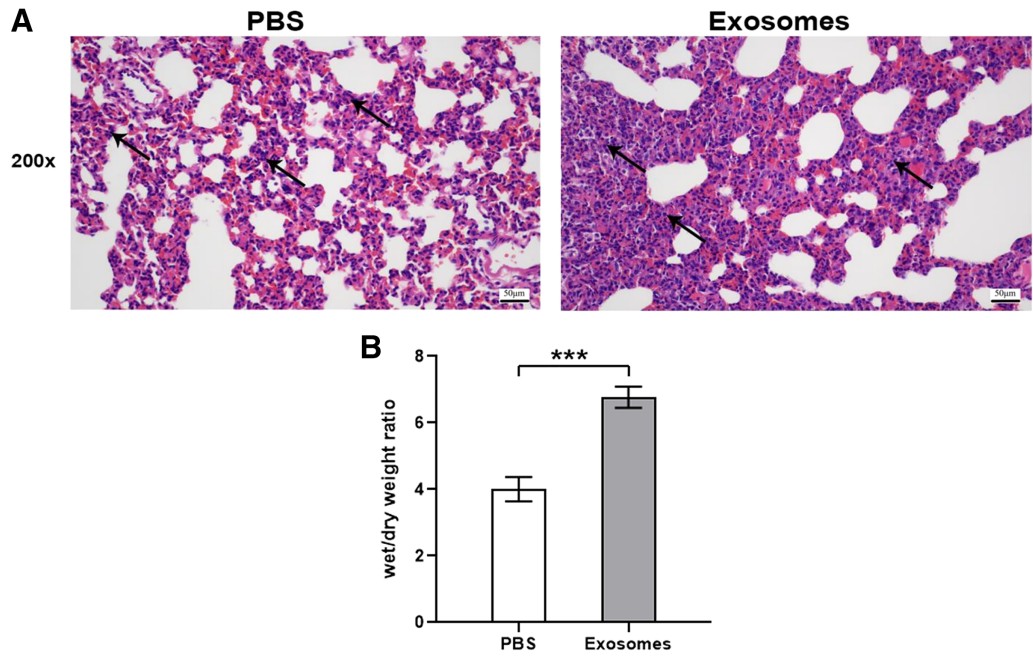

**Figure 2 Histological examination and wet/dry weight ratio of lung tissues of rats injected with ALI exosomes or PBS.** (A) The figure shows that compared with the PBS group, the exosome group had more inflammatory cell infiltration ($n = 5$). (B) The wet and dry lung weight of rats in the exosome group was significantly higher than that in the PBS group. ***$P < 0.001$ ($n = 5$). Scale bar: 50 µm. Magnification: 200×.

led to significant infiltration by neutrophils, with smaller alveolar cavities and full alveolar septum (Fig. 2A). Also, the wet and dry lung weight of rats was significantly higher in the exosome group than in the PBS group (Fig. 2B).

## Exosome confirmation

Nano transmission electron microscopy showed that the diameter of the exosomes, shown as clear vesicle-like structures, was mainly between 40 and 200 nm, primarily around 100 nm; also, they were larger in the ALI group ($n = 2$) (Figs. 3A and 3B). Both exosome surface proteins (CD63 and Tsg101) were shown as positive by Western blot (Fig. 3B). All these findings confirmed the successful extraction of exosomes from BALF.

## High-throughput sequencing results and analysis

The RNA of the exosomes extracted from the BALF in the ALI and control groups was sequenced using high-throughput sequencing (uploaded to NCBI, #SUB7338616). A total of 2,958 differentially expressed lncRNAs were identified, including 2,524 upregulated and 434 downregulated ones. The results were summarized as scatter diagram (Fig. 4A), volcano plot (Fig. 4B), and heatmap (Fig. 4C).

## GO and KEGG database analyses

We conducted an in-depth analysis of the sequencing results. Our analysis showed that there were 5,500 differentially expressed mRNAs between the two groups, of which

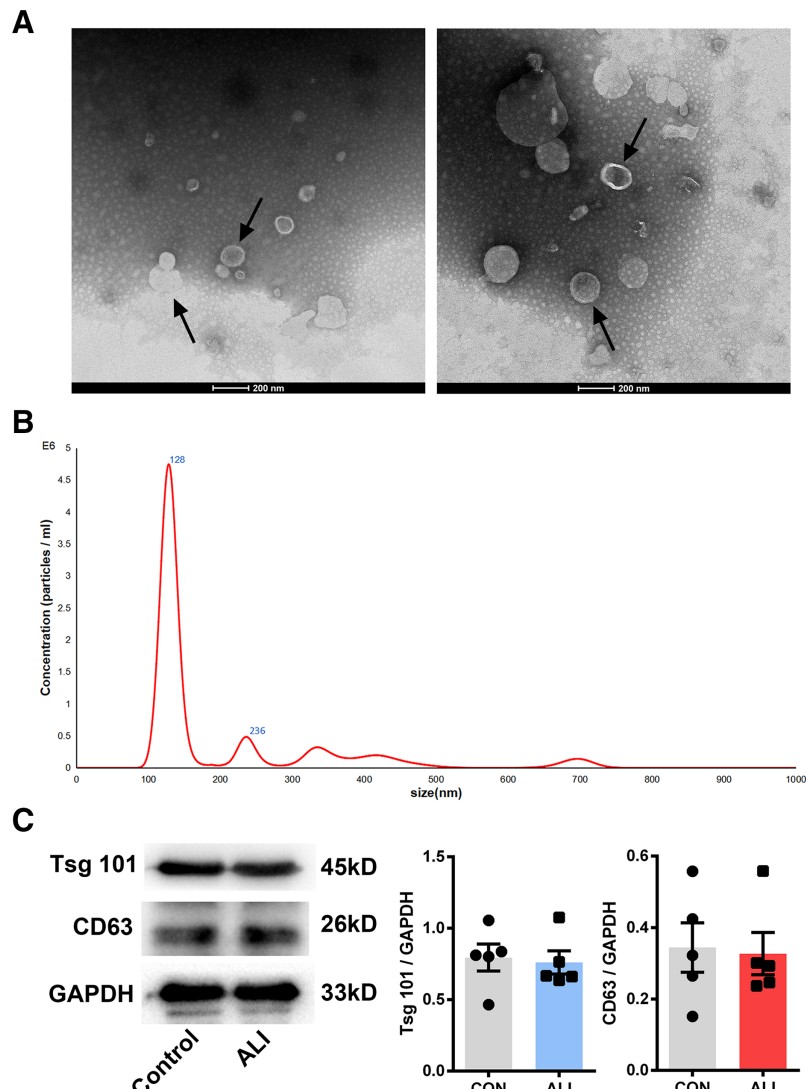

**Figure 3 Validation of exosome characteristics.** (A and B) Transmission electron micrographs of BALF exosomes isolated from the control and ALI rats. Exosomes were found in both control and ALI groups. Scale bar, 200 nm. (C) Western blot analysis of the exosome markers CD63 and Tsg101 in the exosomal preparations. No difference was found in the expression of exosomal markers CD63 and Tsg101 between the control and ALI groups.

2,717 were differentially up-regulated and 2,783 were down-regulated. The mRNAs directly bound to lncRNA and the differentially expressed mRNAs downstream of the differentially expressed lncRNAs (including some novel lncRNAs) were analyzed, and GO and KEGG pathway enrichment analyses were performed on the results. GO enrichment analysis was performed on 2,958 differentially expressed lncRNAs identified. The gene number distribution of top 30 genes in GO analysis is shown in Fig. 5A. As can be seen from the scatter diagram, the three functions with the most significant number of genes included phagocytic vesicle membrane, regulation of receptor biosynthesis process, and I-SMAD binding. Using the same screening criteria as GO analysis, differentially expressed genes for signaling pathways were analyzed using the KEGG database analysis. Salmonella

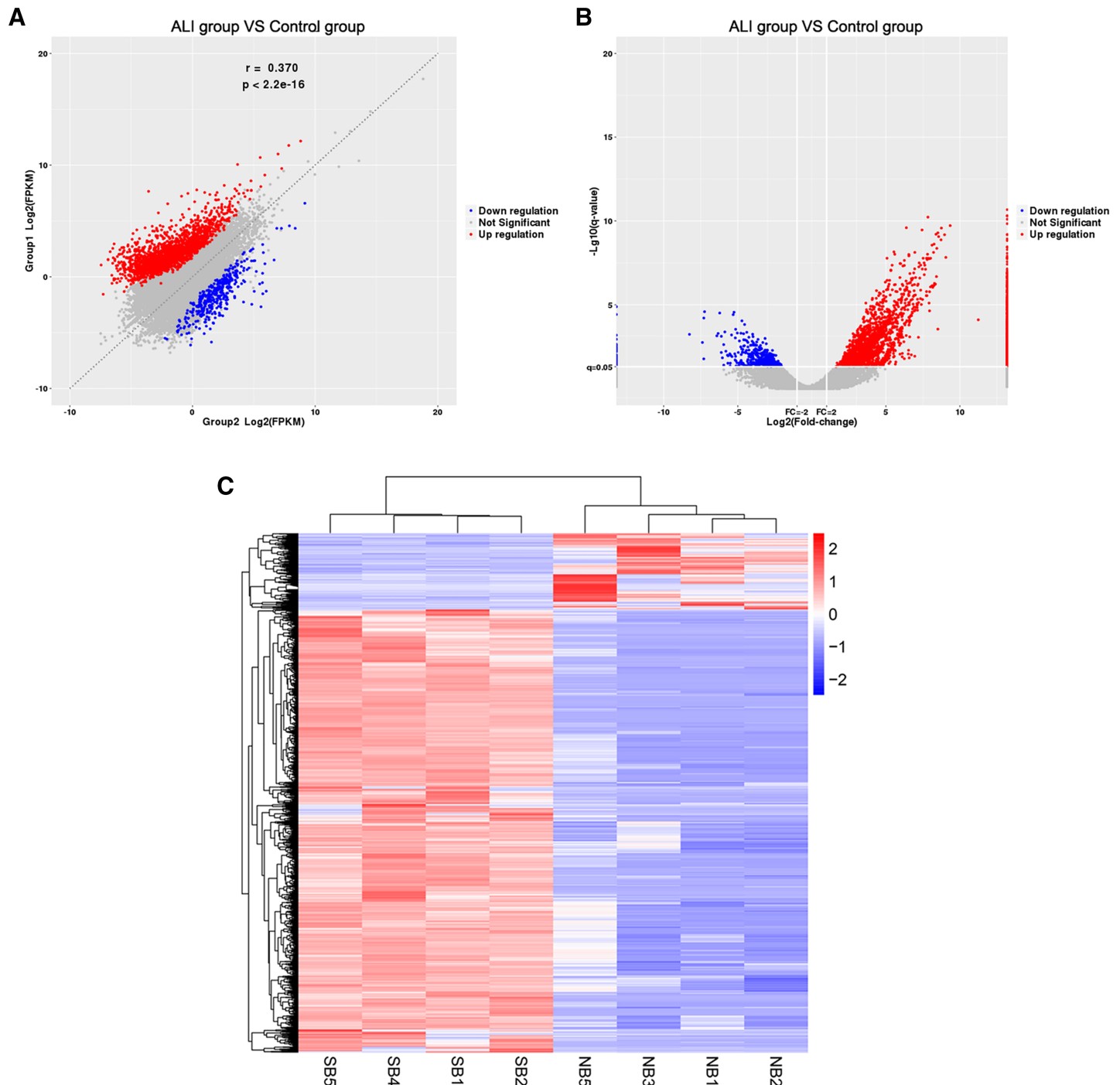

**Figure 4 Exosome lncRNA expression profile in the ALI group compared with the control group.** (A) A scatter plot was used to evaluate the difference in lncRNA expression between the ALI and control groups. (B) Volcano plots. The red points in the plot indicate the upregulated lncRNAs, while the blue points indicate the downregulated lncRNAs. (C) Hierarchical cluster analysis of all lncRNAs differentially expressed in the two groups.

**A**

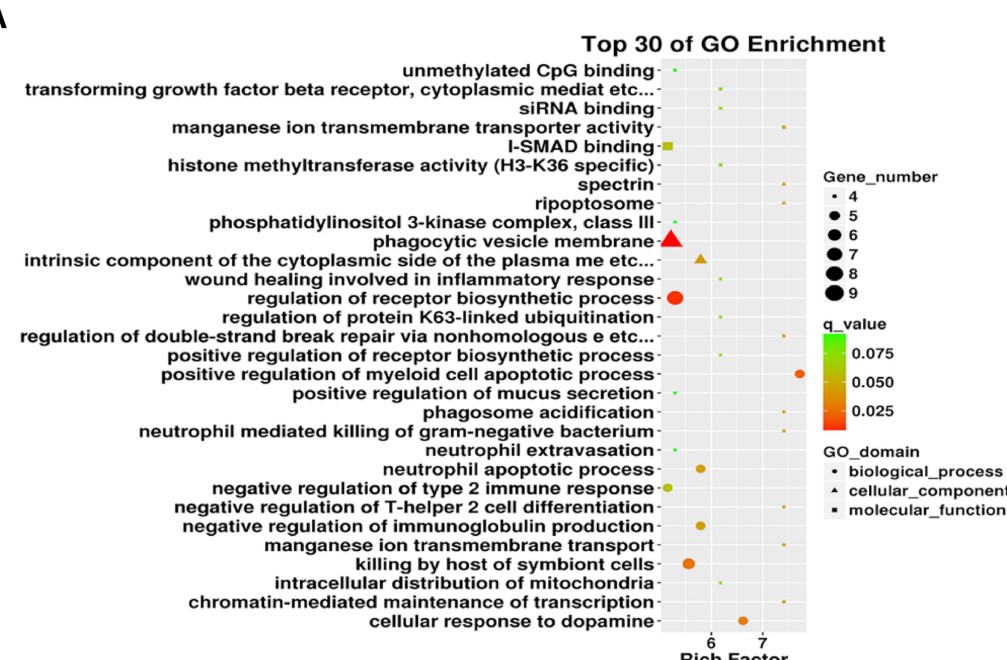

**B**

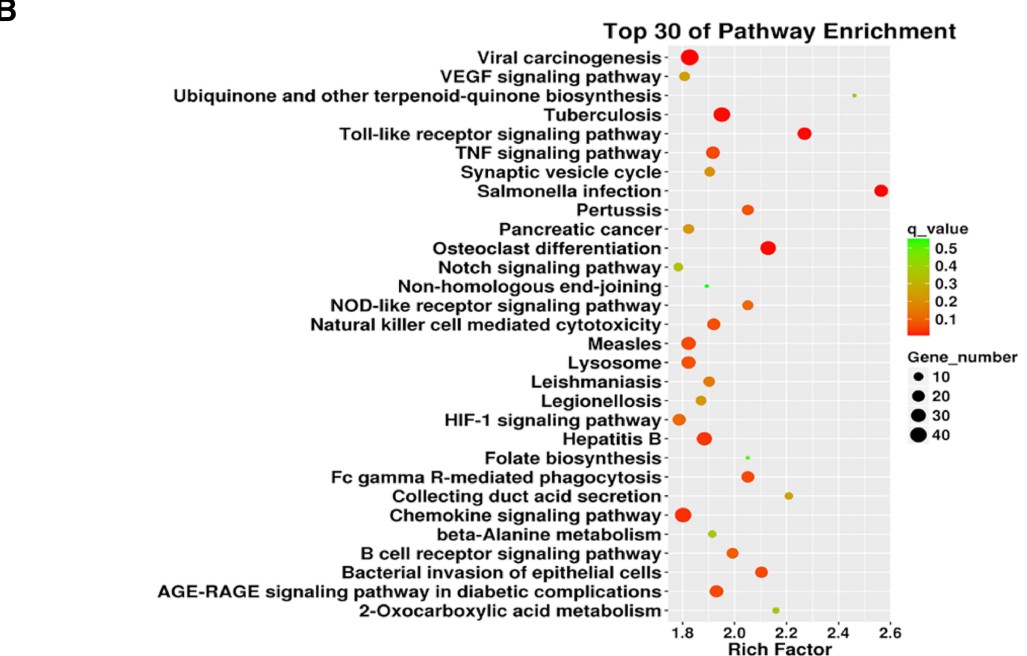

**Figure 5** The 3934 differentially expressed lncRNAs were used for Gene Ontology (GO) enrichment and Kyoto Encyclopedia of Genes and Genomes (KEGG) enrichment analyses. (A) Gene distribution of the top 30 lncRNAs in the GO enrichment. (B) KEGG enrichment of differentially expressed lncRNAs.

infection, Toll-like receptor signaling pathway, and osteoclast differentiation were the most enriched pathways (Fig. 5B). In addition, we list the top 30 differentially up-regulated and differentially down-regulated lncRNAs (Table 2).

**Table 2  Top 30 upregulated and downregulated differentially expressed lncRNAs validated by RNA-seq.**

| lncRNA_id | FPKM (ALI group) | FPKM (Control group) | log2FC | P | Up/down |
|---|---|---|---|---|---|
| MSTRG.34476.1 | 88.335 | 0 | Inf | 1.19E–15 | UP |
| ENSRNOT00000078370 | 50.694 | 0 | Inf | 6.37E–15 | UP |
| MSTRG.18869.2 | 7.647 | 0 | Inf | 1.07E–14 | UP |
| MSTRG.33866.6 | 9.173 | 0 | Inf | 1.11E–14 | UP |
| MSTRG.43407.5 | 14.297 | 0 | Inf | 2.66E–14 | UP |
| MSTRG.17181.2 | 12.371 | 0 | Inf | 5.70E–14 | UP |
| MSTRG.3036.1 | 9.516 | 0 | Inf | 5.03E–13 | UP |
| MSTRG.42423.6 | 13.51 | 0 | Inf | 5.97E–13 | UP |
| MSTRG.6490.3 | 4.249 | 0 | Inf | 1.19E–12 | UP |
| MSTRG.23472.6 | 9.179 | 0 | Inf | 1.66E–12 | UP |
| MSTRG.14472.1 | 3.871 | 0 | Inf | 4.70E–12 | UP |
| MSTRG.27541.7 | 14.782 | 0 | Inf | 5.07E–12 | UP |
| MSTRG.42423.3 | 6.907 | 0 | Inf | 4.96E–12 | UP |
| MSTRG.16199.1 | 18.459 | 0 | Inf | 7.75E–12 | UP |
| MSTRG.18129.1 | 3.107 | 0 | Inf | 1.48E–11 | UP |
| NONRATT021675.2 | 30.011 | 0 | Inf | 2.55E–11 | UP |
| MSTRG.5101.2 | 7.345 | 0 | Inf | 3.07E–11 | UP |
| MSTRG.1751.4 | 3.754 | 0 | Inf | 3.60E–11 | UP |
| MSTRG.27533.5 | 13.525 | 0 | Inf | 3.53E–11 | UP |
| NONRATT021062.2 | 12.975 | 0 | Inf | 3.55E–11 | UP |
| NONRATT026694.2 | 35.768 | 0 | Inf | 4.61E–11 | UP |
| MSTRG.19642.4 | 7.300 | 0 | Inf | 4.74E–11 | UP |
| MSTRG.28850.6 | 1.766 | 0 | Inf | 7.74E–11 | UP |
| MSTRG.14917.1 | 4.494 | 0 | Inf | 1.22E–10 | UP |
| MSTRG.16433.1 | 2.281 | 0 | Inf | 2.62E–10 | UP |
| MSTRG.47785.4 | 4.851 | 0 | Inf | 3.36E–10 | UP |
| MSTRG.18133.1 | 3.184 | 0 | Inf | 4.42E–10 | UP |
| MSTRG.42670.1 | 4.177 | 0 | Inf | 4.41E–10 | UP |
| MSTRG.5096.1 | 3.658 | 0 | Inf | 5.10E–10 | UP |
| NONRATT004688.2 | 14.105 | 0 | Inf | 5.36E–10 | UP |
| NONRATT025820.2 | 0.156 | 47.899 | −8.265 | 2.71E–05 | DOWN |
| MSTRG.25762.7 | 0.017 | 2.91 | −7.388 | 0.00011164 | DOWN |
| MSTRG.14503.11 | 0.41 | 64.435 | −7.297 | 0.001960046 | DOWN |
| MSTRG.30029.2 | 0.181 | 28.421 | −7.291 | 1.55E–06 | DOWN |
| NONRATT025699.2 | 0.349 | 52.885 | −7.243 | 5.25E–07 | DOWN |
| MSTRG.14503.13 | 0.209 | 17.792 | −6.413 | 3.58E–05 | DOWN |
| NONRATT027173.2 | 0.406 | 32.677 | −6.331 | 2.28E–05 | DOWN |
| NONRATT004060.2 | 0.696 | 51.898 | −6.22 | 6.56E–07 | DOWN |
| MSTRG.14503.14 | 0.642 | 44.463 | −6.114 | 0.000626294 | DOWN |
| NONRATT003362.2 | 0.039 | 2.719 | −6.11 | 3.90E–05 | DOWN |
| MSTRG.25762.8 | 0.029 | 1.929 | −6.038 | 0.00163045 | DOWN |

| Table 2 (continued) | | | | | |
|---|---|---|---|---|---|
| lncRNA_id | FPKM (ALI group) | FPKM (Control group) | log2FC | *P* | Up/down |
| NONRATT016515.2 | 0.089 | 5.499 | −5.951 | 0.000162512 | DOWN |
| NONRATT020278.2 | 0.014 | 0.893 | −5.95 | 0.006241881 | DOWN |
| NONRATT002967.2 | 0.056 | 3.283 | −5.875 | 1.75E−05 | DOWN |
| NONRATT002256.2 | 0.057 | 3.199 | −5.8 | 0.000506747 | DOWN |
| NONRATT012252.2 | 0.093 | 4.805 | −5.699 | 0.001011607 | DOWN |
| MSTRG.30218.2 | 0.137 | 6.932 | −5.666 | 0.004342683 | DOWN |
| NONRATT008937.2 | 0.091 | 4.577 | −5.649 | 0.001150031 | DOWN |
| NONRATT021682.2 | 0.047 | 2.172 | −5.536 | 0.000799591 | DOWN |
| NONRATT021161.2 | 0.078 | 3.592 | −5.518 | 0.007494457 | DOWN |
| NONRATT010272.2 | 0.140 | 6.338 | −5.503 | 0.004302169 | DOWN |
| NONRATT028937.2 | 0.021 | 0.945 | −5.470 | 0.001079065 | DOWN |
| NONRATT017373.2 | 0.106 | 4.538 | −5.423 | 3.02E−05 | DOWN |
| NONRATT003609.2 | 1.244 | 52.274 | −5.393 | 0.001310762 | DOWN |
| NONRATT030266.2 | 0.272 | 10.795 | −5.312 | 8.87E−07 | DOWN |
| NONRATT002038.2 | 0.183 | 7.001 | −5.255 | 5.81E−07 | DOWN |
| MSTRG.23080.1 | 0.040 | 1.485 | −5.204 | 0.000798583 | DOWN |
| NONRATT005531.2 | 0.070 | 2.539 | −5.179 | 0.002405774 | DOWN |
| MSTRG.35616.1 | 0.298 | 10.728 | −5.169 | 5.19E−05 | DOWN |
| NONRATT001665.2 | 0.102 | 3.613 | −5.150 | 0.001252922 | DOWN |

## qRT-PCR verification

Of the first 30 differentially expressed lncRNAs, four lncRNAs (NONRATT002967.2, NONRATT003362.2, NONRATT004060.2, NONRATT025040.2, and NONRATT025699.2) starting with NONRATT (*i.e.*, novel lncRNAs) were randomly selected for RT-PCR validation. The PCR validation and sequencing results were consistent ($n = 6$) (Fig. 6).

We analyzed the target genes of NONRATT002967.2, NONRATT004060.2, and NONRATT025699.2 as Tpbg1, Tceb2, and Igf1, respectively. In addition, we also used RT-PCR to verify NONRATT025040.2, whose mechanism will be explored in the future. Its target gene was Foxa1, which was mainly responsible for regulating the differentiation of lung epithelial cells. The results showed that NONRATT025040.2 decreased in the ALI group compared with the control group.

## LncRNA–miRNA interaction networks

Given that lncRNAs can bind to miRNAs and work as a miRNA "sponge", the relation of the five lncRNAs and possible binding miRNAs was investigated. The lncRNA–miRNA interaction network of the five lncRNAs was predicted using miRDB (http://mirdb.org/) and visualized using Cytoscape (Fig. 7). NONRATT004060.2, NONRATT002967.2, NONRATT025699.2, NONRATT025040.2, NONRATT003362.2 interact with 10 miRNAs respectively. NONRATT025699.2 is closely related to NONRATT025040.2, and

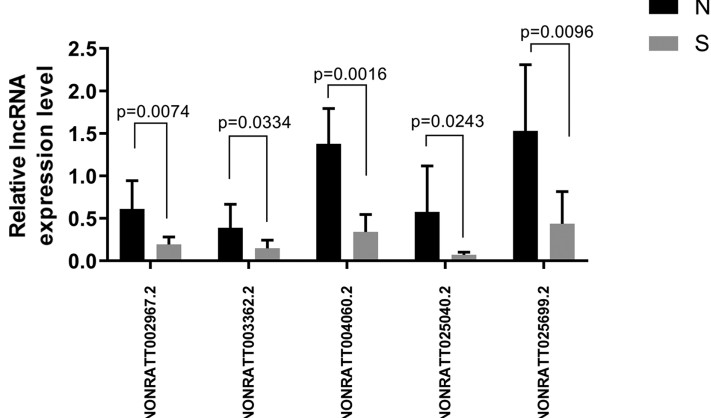

**Figure 6 Five randomly selected differentially expressed lncRNAs between the ALI and control groups were verified by qRT-PCR.** The results showed that the five lncRNA indicators in the ALI group (S) were significantly different compared with the control group (N), $n = 6$, ($P < 0.05$). Data are expressed as the mean ± standard error of the mean.

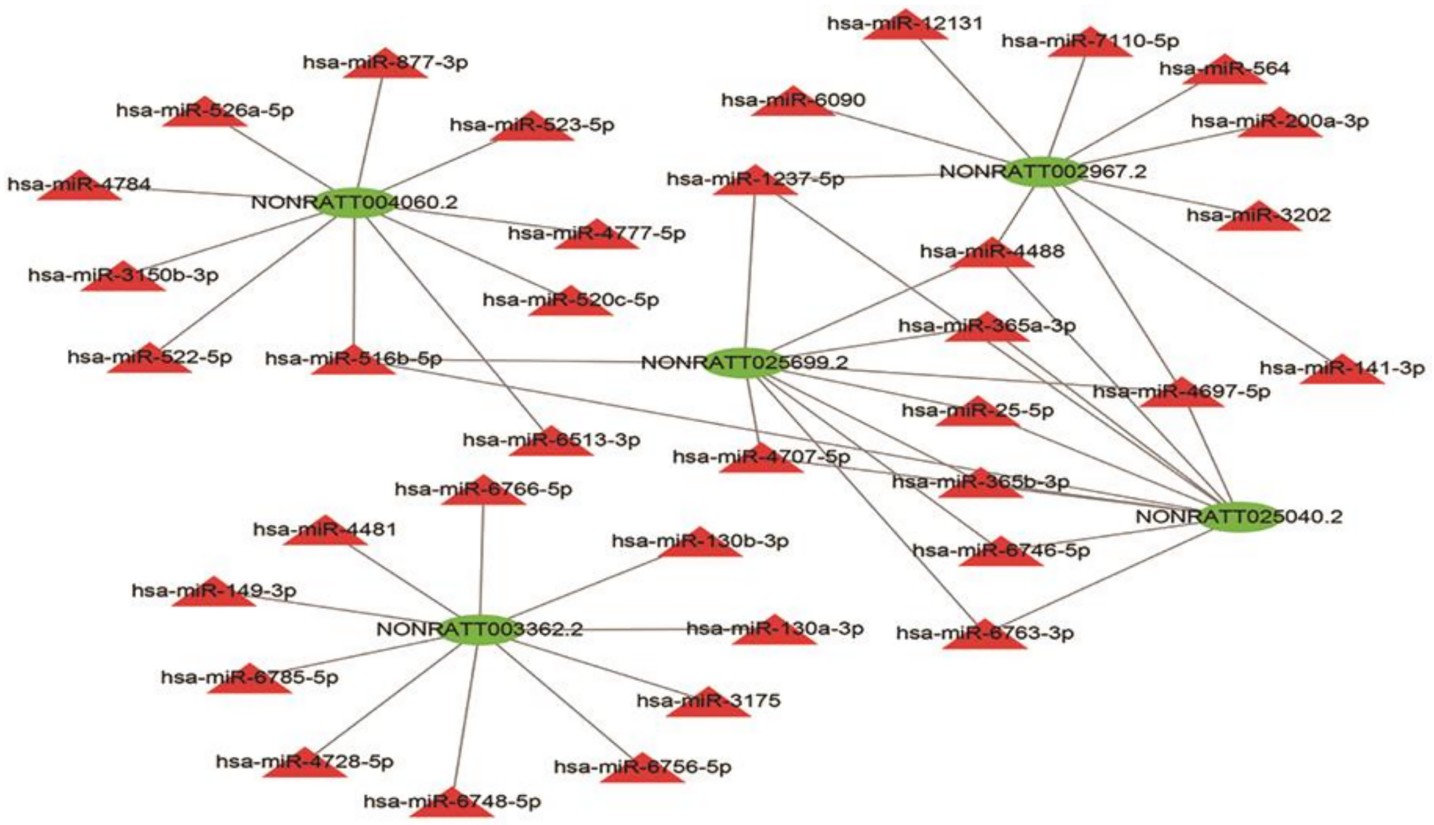

**Figure 7 LncRNA–miRNA interaction network.** A regulatory network for the five validated differentially expressed lncRNAs. Red triangles represent miRNAs. Green triangles represent lncRNAs. The visual display of interaction network of miRNA–mRNA used the Cytoscape software (version 3.7.1, https://cytoscape.org/).

there are eight miRNAs that work together, including hsa-miR-1237-5p, hsa-miR-4488, hsa-miR-365a-3p, hsa-miR-25-5p, hsa-miR-365b-3p, hsa-miR-6746-5p, hsa-miR-6763-3p, hsa-miR-4697-5p. However, NONRATT003362.2 has no interaction relationship with the other four lncRNAs.

## DISCUSSION

BALF can directly reflect the small changes in lung diseases. Also, miRNAs and lncRNAs in BALF exosomes possess different properties according to different underlying diseases (*Lin & Yang, 2018*; *Zhao et al., 2019*). Long noncoding RNAs in different organisms and tissues are different (*Lagarrigue et al., 2021*). Human and animal lncRNA annotations are also different. For example, *Lagarde et al. (2017)* demonstrated GENCODE intergenic lncRNA populations in matched human and mouse tissues for annotation and produced new transcription models of 3,574/561 gene loci, respectively.

LncRNAs in BALFare involved in acute injuries (*Brandenburger et al., 2018*; *Chandran, Mehta & Vemuganti, 2017*), but data are lacking for ALI. Therefore, this study aimed to identify differentially expressed genes in BALF exosomes by RNA sequencing and potential therapeutic targets of ALI.

The present study confirmed that BALF contained exosomes, as first reported by a previous study (*Levänen et al., 2013*). The study also showed that ALI-derived exosomes could induce inflammatory lung changes, as supported by a previous study (*Yuan, Bedi & Sadikot, 2018*). The exosomes mediated crosstalk between cells, contributing to the inflammatory response and structural barrier destruction (*Yuan, Bedi & Sadikot, 2018*). Besides, we used the latest high-throughput sequencing to compare the exosomes in the BALF between the ALI and control groups.

Our research shows 2,958 differentially expressed lncRNAs were identified, including 2,524 upregulated lncRNAs and 434 downregulated lncRNAs, between the ALI and control groups. The top three GO functions were phagocytic vesicle membrane, regulation of receptor biosynthesis process, and I-SMAD binding. Salmonella infection, Toll-like receptor signaling pathway, and osteoclast differentiation were the most enriched KEGG pathways. The GO results showed a considerable number of target genes concentrated in endocytosis, as supported by the reported mechanisms of ALI involving macrophages (*Li et al., 2018*; *Wu et al., 2020*). In addition, the KEGG enrichment analysis showed that most target genes centrally regulated the chemokine signaling pathway. A previous study demonstrated that damaged lung tissues could recruit bone marrow mesenchymal stem cells (*Song et al., 2016*). The recruitment mechanism might be related to the involvement of one or several lncRNAs in exosomes to regulate the chemokine signaling pathway of cells, which needs to be tested. A variety of diseases, including tumors, cardiovascular and cerebrovascular diseases, and diabetes, are multi-gene, multi-factor diseases, and hence it is difficult to achieve an excellent therapeutic effect based on a single target. Possibly a combination of biomarkers for a diagnosis of a disease is a promising approach. *Ware et al. (2010, 2013)* proposed this idea first using eight biomarkers (vWF, SP-D, TNF-R1, IL-6, IL-8, ICAM-1, protein C, and PAI-1) to predict sepsis mortality. A similar approach could be developed for ALI in future studies.

BALF is a better biological fluid than serum or plasma to reflect the overall situation of the lung (*Röpcke et al., 2012*). Furthermore, the application of high-throughput sequencing to detect specific indicators of BALF requires a short time. Defining the specific genes driven by BALF-derived exosomes as a biomarker might improve our understanding of the mechanisms underlying ALI progression, and biomarkers could be derived. Whether a biomarker alone or in combination is more helpful in diagnosing or treating diseases is still controversial. In the clinical setting, various biomarkers, alone or in combination, do not have enough specificity and sensitivity for the diagnosis and monitoring of ALI (*Matute-Bello, Frevert & Martin, 2008*). LncRNAs play a vital role in the biological development of proteins (*Dai et al., 2019*). Still, little is known about the lncRNAs. The main task is to discover more lncRNAs and their biological functions in the future. They may eventually be used as biomarkers for several diseases. Our study provided not only new targets for the diagnosis and treatment of ALI but also new ideas for the diagnosis and treatment of difficult respiratory diseases. Of course, this study was conducted on rats, and the lncRNAs involved were only a superficial exploration. We will select lncRNAs of human and mouse homology and combine the results of GO and KEGG enrichment analyses for related mechanistic research in the future, seeking more target proteins for the diagnosis and treatment of ALI.

# CONCLUSIONS

This study identified differentially expressed lncRNAs in ALI in exosomes from BALF by RNA sequencing. The results showed significant differences in gene expression patterns in ALI-derived exosomes. This study provided a novel theoretical basis for further research on the functions of exosomal lncRNAs in ALI.

# ACKNOWLEDGEMENTS

We thank all the laboratory members for the positive discussions on this subject.

## Funding

This work was supported by the National Natural Science Foundation of China (NSFC) (grant number 81573234); the National Natural Science Foundation of China (NSFC) (grant number 81773445); and the "333" Project of Jiangsu Province (grant number LGY2016006). The funders had no role in study design, data collection and analysis, decision to publish, or preparation of the manuscript.

## Grant Disclosures

The following grant information was disclosed by the authors:
National Natural Science Foundation of China (NSFC): 81573234 and 81773445.
"333" Project of Jiangsu Province: LGY2016006.

## Competing Interests

The authors declare that they have no competing interests.

## Author Contributions

- Meijuan Song conceived and designed the experiments, authored or reviewed drafts of the paper, and approved the final draft.
- Xiuwei Zhang performed the experiments, analyzed the data, prepared figures and/or tables, and approved the final draft.
- Yizhou Gao performed the experiments, prepared figures and/or tables, and approved the final draft.
- Bing Wan conceived and designed the experiments, authored or reviewed drafts of the paper, and approved the final draft.
- Jinqiang Wang performed the experiments, prepared figures and/or tables, and approved the final draft.
- Jinghang Li analyzed the data, prepared figures and/or tables, and approved the final draft.
- Yuanyuan Song analyzed the data, prepared figures and/or tables, and approved the final draft.
- Xiaowei Shen performed the experiments, prepared figures and/or tables, and approved the final draft.
- Li Wang performed the experiments, prepared figures and/or tables, and approved the final draft.
- Mao Huang performed the experiments, analyzed the data, prepared figures and/or tables, and approved the final draft.
- Xiaowei Wang performed the experiments, analyzed the data, prepared figures and/or tables, and approved the final draft.

## Animal Ethics

The following information was supplied relating to ethical approvals (*i.e.*, approving body and any reference numbers):

The experiments were performed following institutional guidelines and approved protocols. The animal experiments were approved by the Institutional Animal Care and Use Committee of Nanjing Medical University (No. IACUC-2004021).

## DNA Deposition

The following information was supplied regarding the deposition of DNA sequences:

The high-throughput sequencing data is available at NCBI: https://www.ncbi.nlm.nih.gov/guide/sequence-analysis/.

The Control Group is available at NCBI: NB1: SRR11892823; NB2: SRR11892822; NB3: SRR11892821; NB5: SRR11892820.

The Test Group is available at NCBI: SB1: SRR11892819; SB2: SRR11892826; SB3: SRR11892825; SB5: SRR11892824.

## Data Availability

The raw data are available in the Supplemental Files.

## Supplemental Information

Supplemental information for this article can be found online at http://dx.doi.org/10.7717/peerj.13159#supplemental-information.

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
