# Peer review of "RNA sequencing reveals the emerging role of bronchoalveolar lavage fluid exosome lncRNAs in acute lung injury"

_PeerJ, doi:10.7717/peerj.13159_

## Round 0.1 · original submission · Major Revisions

The reviewers have all provided a clear and insightful analysis of your paper. All recognise it has merits, but all identify issues that must be fully and carefully addressed prior to publication. All points must be addressed; this may require further experimental analysis.

Exosomes and EV isolation methods do not appear to conform to the ISEV minimum standards. Hence, there is a major concern that the data lack robustness. The details of the methods used need to be provided with reference to the ISEV standards, and there is a clear caveat that if the methods do not conform then the paper cannot be accepted as the underlying premise may be flawed.

Why has the RNAseq identified no protein-encoding mRNA? None of this is reported. In figure 2, there should be more detailed analysis of the pRT-PCR data. The cytoscape image is poor. Consider the points regarding the choice of exemplar species to analyse.

Please note the comments of all reviewers and address all comments. This may require more experimental work and will definitely require further data analysis.

Respond to ALL points in detail and explain your reasoning for all comments made in response to the reviewers. You should also work please on the presentation of the paper, as all reviewers have identified confusion arising from phraseology and writing.

·

Basic reporting

1. Overall, this article was written clearly and professionally, however, some text may be ambiguous or unclear.
ie:
a. The second sentence of background in the abstract (line 31-32) was grammarly incorrect.
b. Some method was not explained clearly. For example, the sex of control mice (line 34), how exosomes were obtained (line 35), and what does it mean that nothing was given to control group mice (line 117).
c. Some abbreviations were not explained at first use, ie LPS.

2. The article has included an introduction from a wide perspective to explain the background of the work and demonstrate how this work contributed to the field. However, it would be better if the author could explain: why do you select exosomes instead of other types of extracellular vesicles (EV) in the study, what was the significance of exosomes compared with other types of EV, ie MV and apoptosis bodies.

3. This article was structured properly with an acceptable format of 'standard sections'. All appropriate raw data was supplied. Figures and tables were relevant to the content of the article of sufficient resolution.
However, it would be helpful if more descriptive information could be explained in the captions of the figures and tables to allow them to explain themselves independently.
For example,
a. Sample size, sex of mice, etc in each figure/ table.
b. Use of abbreviations, ie FPKM (table 2), log2FC (table 2).
c. This might not be your fault, but the title of table 1 was not in the correct place.

Overall, the current study was represented as an appropriate unit of publication and include necessary results relevant to the hypothesis.

Experimental design

This is original primary research within the aims and scope of the journal.

It has clearly defined the research question relevantly and meaningfully. The knowledge gap has been defined and being investigated, and the results have contributed to filling that gap.

The investigation was at a high technical standard, ie exosomes sizing, miRNA profiling, etc. The research was conducted in conformity with the prevailing ethical standards in the field.

The method of the current study was described with sufficient detail and information to replicate.

Validity of the findings

1. The current study gave an outcome based on the results, the findings were novel and meaningful.


2. Most necessary results were supplied. However, it would be better if you could show how the raw data from RNA-sequencing was normalised and validated (prior to table 2).

3. Conclusions were well stated. However, in the discussion, it would be better if you could talk further and deeper in: to what extent do lncRNA were the same/ different between exosomes and other resources, and between humans and mice. Also, in conclusion, you should emphasise that these findings were based on mice study rather than human.

Additional comments

Overall, this research was novel and meaningful, however, minor revisions should be made.

·

Basic reporting

The English language should be improved for a better flow of the story. For example, sentence structure errors in lines 31, the lack of a full stop at the end of line 120, choice of words (i.e. bottom) in lines 151 and 152 makes comprehension challenging.

The main figures should not contain sentences or paragraphs since they are already in the figure legends. In my opinion, the order by which Figures 2 and 3 are presented should be reversed for a better flow, i.e. the current Figure 3 (exosome characterization) should be Figure 2 and the current Figure 2 (histological examination of the lung tissue upon exosome injection) should be Figure 3.

For Figure 6, what does the panel legend (N and S) represent? Do the authors mean "control vs ALI group"?

The authors discussed about the top 30 differentially expressed lncRNAs between the ALI and control group but only provided the top 20 lncRNAs in Table 2.

In Table 2 under the column of "log2FC", what does "Inf" represent? The authors should provide clear elaborations for any acronyms used in the manuscript.

Experimental design

The authors mentioned in line 117 that nothing was given to the rats in the control group. In my opinion, the authors should repeat the experiment with PBS solution for the airway injection for the control groups so as to eliminate any possibility that the lung injury could be induced during the injection process, rather than the LPS.

The authors "randomly" selected 5 of the top 30 differentially expressed lncRNAs for qRT_PCR validation. In my opinion, it is more biologically meaningful if the authors could provide a rationale for the selection process, or select the top 5 most dysregulated lncRNAs of a particular important KEGG or GO pathway for qPCR validation, or simply the top 5 most differentially expressed lncRNAs for qPCR validation so as to provide the basis for future studies.

Validity of the findings

The authors concluded that there was an increased neutrophil infiltration (line 230-231) in the lung tissues of the ALI group compared to control, but there is no quantitative measurement to convincingly demonstrate that. The authors should perform via for example, flow cytometry, to quantitatively demonstrate the % neutrophil infiltration and examine any differences between the two groups.

Additional comments

Major comments

1) For the current Figure 2 whereby the authors demonstrated the histological differences between rats injected with PBS or ALI-derived exosomes, more quantitative assays should be performed just as in Figure 1, i.e. Wet-dry ratio, neutrophil infiltration, cytokine secretion.

2) While characterizing the exosomes isolated from the BALF of the rats (Figure 3), the authors should also checked for any differences in the abundance/quantity of exosomes between the ALI and control groups since it has been reported that LPS (infectious stimuli) can lead to an increase in EV secretion.

3) While the authors provided the lncRNA-miRNA regulatory network predictions for the 5 validated lncRNAs, it may be more biologically meaningful to compare any overlaps of the network between the 5 lncRNAs or provide a merged network for some of the differentially expressed lncRNAs of the same GO or KEGG pathway. This would provide a stronger basis for future experimental studies to characterize these lncRNAs and their potential diagnostic and therapeutic roles in ALI.

Reviewer 3 ·

Basic reporting

BASIC REPORTING
a. Clear, unambiguous and professional language? See notes
b. Intro shows context? Relevant background and context is presented but there are key pieces of information missing
c. Well referenced and relevant literature? Mostly, there is one or two occasions when reference is required but missing
d. Structure conforms to peerJ standards? As far as I am aware, yes
e. Figures are relevant and high quality, well labelled and well described? Figures are well-described, some specific comments below.
f. Raw data supplied? Raw data availability is clearly stated in methods

• Lines 31-32 in abstract, missing contextualisation for comprehension - currently statement is the subordinate clause of a complex sentence
• Line 34 Undefined acronyms in abstracts (for example LPS)
• Lines 69-72
• Lines 82-83, 127-29 appear disjointed from the flow of information presented around it
• General disjointed flow of information, often there is too little detail more is needed (for example the explanation of EV see below) and too much information when it is not required (for example lines 104-120 in methods could be made more concise and irrelevant information could be removed).
• Language used could also be improved to better reflect norms of the English writing, for example:
• lines 215-216 uses "the" when it could be removed and improve general flow of writing
• Line 239 the plural were should be used over singular was
• Line 266 the plural "sponges" or addition of "an" before the noun
• Subheadings in results section could be improved for better precision and clarity in understanding
• Results are concise and well-focused in general. The final subheading (lines 266-269) does not describe any results - the language is methodological. The key aspects of the figure could be described.
• Use of inappropriate adverbs at the beginning of sentences in the discussion decreases the clarity of information
• Figure 3 B; if scale bar is correct, there is an arrow idiciating an exosome pointing to a structure many magnitudes too large

Experimental design

• Aim of the study could be presented a more clear and logical manner, no clear hypothesis is stated.
• More detail or improved language in the description of control animals - were they subjected to anaesthesia even though controls did not receive a control saline injection?
• Lines 133-134 do not clearly convey method
• At the moment, the reporting of EV isolation and characterisation does not conform to ISEV minimum reporting standards. The description of EV isolation lacks reporting centrifugal forces, models of centrifuge, volumes of starting material etc. reference to ISEV minimum reporting standards could be included.
• Authors give no explanation for mapping rat RNA sequencing results to the human genome assembly?
• Authors state that an ALI standard was used to exclude data without definition or reference to source
• Authors clearly state the availability of data but do not make comment on the quality of the RNA sequenced

Validity of the findings

• Both the conclusions and aims, whilst well stated, do not accurately reflect the detail of the experiments presented and are more conceptual. The characteristics of BALF-derived exosomes from ALI and control is described and compared with key differences. Potential functions and interactions with miRNA are explored but not experimentally validated. Therefore the conclusion that BALF-derived exosomes play an important role in the development of ALI is perhaps overreaching the scope of the experimental design.
• The discussion appears to jump between analysis of the pathways or mechanisms involved in progression of ALI and applicability of findings as biomarkers, this can make some sections of the discussion confusing and disjointed. The consideration of BALF as a clinically accessible and viable source for biomarker detection is good.
• Both the conclusions and aims, whilst well stated, do not accurately reflect the detail of the experiments presented and are more conceptual. The characteristics of BALF-derived exosomes from ALI and control is described and compared with key differences. Potential functions and interactions with miRNA are explored but not experimentally validated. Therefore the conclusion that BALF-derived exosomes play an important role in the development of ALI is perhaps overreaching the scope of the experimental design.
• The discussion appears to jump between analysis of the pathways or mechanisms involved in progression of ALI and applicability of findings as biomarkers, this can make some sections of the discussion confusing and disjointed. The consideration of BALF as a clinically accessible and viable source for biomarker detection is good.

Additional comments

• Exosomes and EVs are poorly defined and differentiated from one another in first presentation. Role of exosomes as messengers between relevant cell types is not clearly defined or explained during early introduction.
• The mapping of rat-derived genetic information to the human genome is never justified, discussed or explained however the lncRNA id's are listed as rat IDs? Is the methodology reported correctly? miRDB names are reported as human miRNAs.
• Figure 1, can all the data points be added to the graph as in figure 3c.
• In Figure 2, please add the wet/ dry data to allow comparison with figure 1b.
• Figure 3 is only a snap shot of the size, if nanoparticle data tracking is available, this data should be presented as it gives a more representative indication of particle size.
• What was the result of the RNAseq analysis, as well as the identification of significantly differentially regulated lncRNAs, what other species of RNA were identified and differentially regulated?
• What is the evidence that the selected lncRNAs act as miRNA sponges, is there any evidence the lncRNAs are they implicated in other functions and disease?
• Can the authors provide the detailed analysis data to the reviewers, for the qRT-PCR, Ct and delta Ct information. Were the lnc RNA Taqman probes commercially available or custom generated?
• Can the authors comment on the on the reasons for using the polyA+ set of RNA as opposed to all long RNAs?
• Is there any evidence that the cytoscape figure is enriched for miRNAs implicated in the pathways highlighted in Figure 5?

---

## Round 0.2 · Minor Revisions

Thanks for the rebuttal. A few minor points remain:

Why has the RNAseq identified no protein-encoding mRNA? None of this is reported.

Line 432 – how do you define ‘excellent’ for sequencing reads? Please expand this.

Please specify numbers of technical and biological replicates in figures 1 and 6.

Direct citation of the relevant ISEV protocol should be added.

·

Basic reporting

Please ensure that the added text and figures/figure legends are well read through for the English grammar and structure.

Experimental design

No comment

Validity of the findings

No comment

---

## Round 0.3 · accepted · Accept

Thanks for attending to the remaining clarifications. Congratulations on an interesting study.